# Design, Optimization and Validation of the ARMS PCR Protocol for the Rapid Diagnosis of Wilson’s Disease Using a Panel of 14 Common Mutations for the European Population

**DOI:** 10.3390/genes13111940

**Published:** 2022-10-25

**Authors:** Mikhail Maksimovich Garbuz, Anna Alexandrovna Ovchinnikova, Vadim Vladimirovich Kumeiko

**Affiliations:** 1Institute of Life Sciences and Biomedicine, School of Natural Sciences, Far Eastern Federal University, Vladivostok 690922, Russia; garbuzmihail.93@gmail.com (M.M.G.); miha.rs@mail.ru (A.A.O.); 2A.V. Zhirmunsky National Scientific Center of Marine Biology, Far Eastern Branch of Russian Academy of Sciences, Federal University, Vladivostok 690041, Russia

**Keywords:** Wilson’s disease, WD, ARMS

## Abstract

Background: Wilson’s disease (WD) is an autosomal recessive inherited disorder of copper metabolism resulting from various mutations in the *ATP7B* gene. Despite good knowledge and successful treatment options, WD is a severe disease that leads to disability, destructively affecting the quality of life of patients. Currently, none of the available laboratory tests can be considered universal and specific for the diagnosis of WD. Therefore, the introduction of genetic diagnostic methods that allow for the identification of the root cause at any stage over the course of the disease gave hope for an earlier solution of diagnostic issues in patients with WD. Methods: A method for the genetic diagnosis of WD based on ARMS PCR, DreamTaq Green PCR Master Mix and modified primers has been developed. This method is able to detect 14 mutant alleles: p.His1069Gln, p.Glu1064Lys, p.Met769HisfsTer26, p.Gly710Ser, p.Ser744Pro, p.Ala1135GlnfsTer13, p.Arg778Leu, p.Arg1041Trp, p.Arg616Gln, p.Arg778Gly, p.Trp779*, p.Val834Asp, p.Gly943Ser and p.3222_3243+21del43. Results: The primers for all mutations were highly specific with an absence of wild-type amplification. All the results were validated by direct DNA Sanger sequencing. Conclusions: This fast and economical method provides coverage for the identified common mutations, thereby making ARMS PCR analysis using DreamTaq Green PCR Master Mix and modified primers feasible and attractive for large-scale routine use.

## 1. Introduction

Wilson’s disease (WD) (MIM 277900) is an autosomal recessive disorder of copper metabolism resulting from various mutations in the *ATP7B* gene. This gene consists of 21 exons and encodes the protein ATPase, which is involved both in the release of excess copper into bile and in the incorporation of copper into apocaeroloplasmin for the synthesis of functional ceruloplasmin [1,2]. The disease manifests itself due to the gradual accumulation of copper in the tissues of the liver and brain, followed by cell death, tissue damage, the release of excess free copper and its deposition in other organs [3]. It is one of the first genetic diseases for which an effective pharmacological treatment has been established. WD was previously thought to be a rare disease, but studies in several different regions have shown that the occurrence of this disease depends on the population and may occur in more than an estimated 1 in every 35,000–45,000 people [3]. The sequencing of the coding region of the *ATP7B* gene and adjacent splice sites in 1000 newborns revealed a heterozygous frequency of carriers of the *ATP7B* mutation of 1 in 40, which is in connection with the incidence in the UK—which was calculated as 1 in 7026 people [4].

The main problem associated with WD is diagnosis and accurate diagnosis. Patients with WD have a wide range of symptoms associated with various organs. The most common symptoms are liver dysfunction, neuropsychiatric disorders, Kaiser–Fleischer rings on the cornea, and hemolysis caused by acute liver failure. The symptoms of WD can appear at any age [5]. An accurate diagnosis is established after a thorough assessment of the clinical picture, genetic and biochemical tests, as well as after detecting a violation of copper metabolism. However, even in this case, one should carefully evaluate the data and adjust the results by taking into account the patient’s age, habits, lifestyle, while not ruling out the possibility of similar diseases [3].

The diagnosis of WD is based on the results of clinical, biochemical, histological and genetic tests and parameters. To assess this disease, the Leipzig criteria were established to help standardize the diagnosis and treatment of WD. The key clinical diagnostic features that are used to form the Leipzig criteria are liver disease, motor and neuropsychiatric disorders, Kaiser–Fleischer (KF) rings on the cornea, and acute hemolysis caused by acute liver failure [6,7]. However, establishing an accurate diagnosis based on the results of the examination of the patient and biochemical parameters takes a long time, during which irreversible changes may occur in the patient’s body, or he may die without proper treatment. On the contrary, the time saved with the help of molecular genetic diagnostics is significant.

Direct molecular genetic diagnosis is difficult due to the 800 possible mutations that can be encountered (https://www.ualberta.ca/medical-genetics/index.html, accessed on 5 June 2022). Several frequent mutations show a peculiar global distribution [6]. In addition, most patients are compound heterozygotes (that is, they carry two different mutations) [8,9,10]. The general mutation screening method is useful in populations where the founder effect occurs at a high frequency. The p.H1069Q mutation is widespread in Caucasians with 26–70% of all alleles [6,11], while p.R778L accounts for up to 44% of frequencies in East Asia [12,13,14,15,16]. Studies of the western part of Russia using NGS among more than 400 unrelated probands revealed 66 pathogenic alleles, wherein the most frequent mutations were p.His1069Gln (50%), p.Met769HisfsTer26 (5.3%) and p.Ala1135GlnfsTer13 (2.6%) [17]. However, historically, the territory of Russia was formed from a large number of nationalities from different regions, which undoubtedly affected the spectrum of mutations in the *ATP7B* gene.

The p.His1069Gln mutation is the most common in Central and Eastern Europe, and therefore it has been suggested that this mutation originated in Eastern Europe [18]. The occurrence of this mutation in European countries in ascending order is observed in France (15%) [8], Turkey (17.4%) [19], continental Italy (17.5%) [20], Denmark (18%) [21], Great Britain (19%) [4], Netherlands (33%) [22], Austria (34.1%) [6], Greece (35%) [23], Sweden (38%) [24], Romania (38.1%) [25], Hungary (42.9%) [18], former Western Germany (47.9%) [26], former Yugoslavia (Slovenia, Croatia, Bosnia and Herzegovina, Serbia, Montenegro, Macedonia) (48.9%) [27], Czech Republic and Slovakia (57%) [28], Bulgaria (55.8%) [29], former East Germany (63%) [26], and the highest occurrence is mentioned in Poland (72%) [30]. South of the Alps, this mutation becomes infrequent and is completely absent in Sardinia [31]. Over the course of our study, 100 DNA samples from patients from the Far East of Russia with suspected WD were sequenced, and an economical and rapid method for diagnosing this disease was developed based on the identified mutations. The method was based on ARMS PCR, DreamTaq Green PCR Master Mix and modified primers. This method is able to detect 14 mutant alleles: p.His1069Gln, p.Glu1064Lys, p.Met769HisfsTer26, p.Gly710Ser, p.Ser744Pro, p.Ala1135GlnfsTer13, p.Arg778Leu, p.Arg1041Trp, p.Arg616Gln, p.Arg778Gly, p.Trp779*, p.Val834Asp and p.Gly943Ser и с.3222_3243+21del43.

## 2. Materials and Methods

### 2.1. Research Materials

The materials for the study were 54 patients with a previously established diagnosis of VD, 14 patients with suspected presence of this disease and 32 first-degree relatives (a total of 100 people). Patients for genetic testing were residents of Primorsky Krai and the Far East, regardless of gender, age and nationality. The selection of patients took place in connection with the treatment of complaints of problems of a behavioral and neurological nature. The collection of material took place from 2020–2022 at the Nevron MC and the Health MC after the diagnosis was made. The examination group included both patients with a preliminary diagnosis as well as long-term follow-up and therapy patients with an established diagnosis of WD.

Blood samples were obtained only from patients who signed up for the study participation information form. The use of human tissue material and animal experiments involving rats were approved by the FEFU Ethics Committee according to Resolution #5/19 December 2017.

### 2.2. Methods

#### 2.2.1. Isolation of Genomic DNA

DNA extraction from whole blood samples was carried out using the ExtractDNA Blood reagent kit for isolation and purification of genomic DNA from whole blood (Evrogen, Moscow, Russia) according to the manufacturer’s protocols.

#### 2.2.2. Polymerase Chain Reaction Method

Before sequencing the *ATP7B* gene fragments according to Sanger, the fragments were amplified. The polymerase chain reaction (PCR) method was used to amplify certain DNA fragments. The reaction mixture for amplification of genomic DNA (20 μL) included: deionized water—5.5 μL; DreamTaq Green PCR Master Mix (Thermo Scientific™, Waltham, MA, USA)—10 μL; primers (10 pmol/μL)—1 μL each; DNA—2.5 μL. PCR for genomic DNA was carried out under the following conditions: preliminary denaturation at 95 °С—2 min; further for 35 cycles: 95 °С denaturation—30 s, 63 °С annealing—20 s, 72 °С synthesis—40 s; final cycle 72 °С—2 min.

The nucleotide sequences of the gene-specific primers used are shown in Table 1.

Results were visualized by 1.5% agarose gel electrophoresis with ethidium bromide (10 mg/mL, Evrogen) in TAE buffer (40 mM Tris-acetate buffer, pH 7.6, 1 mM EDTA) and photographed on ChemiDoc MP Imaging System (BioRad, Hercules, CA, USA) in transmitted ultraviolet light. To determine the length of the fragments, a DNA marker of 1000 base pairs (bp) (Evrogen) was used.

#### 2.2.3. Purification of PCR Fragments

Purification of the PCR mixture before the BigDye reaction was carried out by precipitation with sodium acetate (C2H3O2Na). A total of 1.8 μL of sodium acetate was added to the PCR mixture, then 54 μL of 96% ethanol was added and left at −80 °C for 5 min. After the expiration of time, centrifugation was carried out at +4 °С at a speed of 13.4 thousand rpm for 30 min. After centrifugation, the supernatant was decanted and 100 μL of 70% ethanol was added and centrifuged again at +4 °C at a speed of 13.4 thousand rpm for 30 min. The supernatant was then poured off again and left to dry at room temperature for 15 minutes. At the end of the drying time, it was eluted in 20 μL of deionized water.

#### 2.2.4. Sanger Direct Sequencing Method

The determination of the nucleotide sequence was carried out by the classical method of direct automatic Sanger sequencing of the purified PCR product from direct primers. The reaction mixture for carrying out the BigDye reaction (10 μL) included: deionized water—6.25 μL; BigDye Terminator v3.1 (Thermo Scientific™)—0.5 μL; BigDye Terminator v3.1 Sequencing Buffer 5X (Thermo Scientific™)—1.75 μL; forward primer (5 pmol/μL)—0.5 μL; DNA—1 μL. The forward primers were taken from Table 1. The BigDye reaction was carried out under the following conditions: pre-denaturation 96 °C—1 min; further for 25 cycles: 96 °С denaturation—10 s, 63 °С annealing—20 s, 60 °С synthesis—4 min.

#### 2.2.5. Cleaning the BigDye Mixture

Purification of the BigDye mixture was performed with EDTA. A total of 2.5 μL of 0.125 M EDTA (pH 8.0) and 55 μL of 96% ethanol were added to the reaction mixture, after which it was left for 20 min at −20 °C. At the end of the time, the mixture was centrifuged at a speed of 13.4 thousand rpm at +4 °C for 30 min. After centrifugation, the supernatant was decanted and 100 μL of 70% ethanol was added and centrifuged again at +4 °C at a speed of 13.4 thousand rpm for 25 min. The supernatant was then poured off again and left to dry at room temperature for 8 min. Dried samples were dissolved in 35 μL of formamide (Thermo Scientific™) and stored at −20 °C until sequencing.

#### 2.2.6. Sequencing and Analysis of Obtained Data

The nucleotide sequence was determined by direct automatic Sanger sequencing of a purified PCR product from a forward primer. Sequencing was performed according to the manufacturer’s protocol on an ABI Prism 3100 instrument (Applied Biosystems, Waltham, MA, USA).

Analysis of the resulting nucleotide sequences was performed using the Vector NTI Advance 9.1.0 (Invitrogen, Waltham, MA, USA) and MEGA-X (MEGA Software, State College, PA, USA) programs. The search for homologous sequences for analysis was performed using the BLAST server (http://blast.ncbi.nlm.nih.gov, accessed on 1 September 2022). Multiple alignment of nucleotide sequences and their analysis was performed using the MEGA-X software.

#### 2.2.7. Allele-Specific PCR Method and its Modification

Allele-specific PCR is based on the use of allele-specific primers selected strictly for mutation sites. To implement the technology, two versions of universal primers are used: the first option is when the primer must be strictly complementary in its 3′-terminal nucleotide, which corresponds to the nucleotide of the template DNA. The second option is when the universal primer contains a 3′-terminal nucleotide, which is always non-complementary to the template, and the mutant nucleotide of the template falls into its inner part. Such primers allow for the detection of any point mutations (SNPs) in the homozygous state. To identify the heterozygous state of the analyzed gene, “mutant” and “normal” primers can be used in two different tubes. SNPdetect polymerase is used for this test.

Our modification consists of designing unique primers that are modified with one nucleotide mismatch at the 3′ end in addition to the mismatch in the mutant primer. Modification occurs according to the rule of strong and weak nucleotide mismatches: strong mismatches are C-C, G-A and A-A; and weak mismatches are T-T, T-C, T-G, G-G and A-C [32]. The nucleotide sequences of the gene-specific primers used are shown in Table 2.

The reaction mixture for allele-specific PCR (20 μL) included: deionized water—5.5 μL; DreamTaq Green PCR Master Mix (Thermo Scientific™)—10 μL primers (10 pmol/μL)—1 μL each; DNA—2.5 μL. PCR for allele-specific PCR was carried out under the following conditions: preliminary denaturation 95 °С—5 min; further for 35 cycles: 95 °С denaturation—30 s, 65 °С annealing—30 s, 72 °С synthesis—40 s; final cycle 72 °С—3 min.

Results were visualized by 1.5% agarose gel electrophoresis with ethidium bromide (10 mg/mL, Evrogen) in TAE buffer (40 mM Tris-acetate buffer, pH 7.6, 1 mM EDTA) and photographed on a ChemiDoc MP Imaging System (BioRad) in transmitted ultraviolet light. To determine the length of the fragments, a DNA marker of 1000 base pairs (bp) (Evrogen) was used.

## 3. Results

Sanger sequencing of all samples was performed in this study as a gold standard and for the comparison of the obtained data. Based on the results from sequencing the genetic material of 54 patients with a previously established diagnosis of WD, 14 patients with suspected presence of this disease and 32 first-degree relatives (100 people in total), 6 people were identified without mutations in the *ATP7B* gene, 27 patients were identified with heterozygous carriage and 67 patients were identified with a pathogenic mutation. Sanger sequencing revealed that the most common mutation (48%) is p.His1069Gln (c.3207C > A), located in exon 14 of the *ATP7B* gene. Mutations p.Glu1064Lys (c.3190G > A) (20%) and p.Met769HisfsTer26 (c.2304insC) (8%) of exons 14 and 8, respectively, were also common. In 4% of cases, p.Gly710Ser (c.2128G > A) and p.Ser744Pro (c.2230T > C) mutations located on exon 8 and a p.Ala1135GlnfsTer13 (c.3402delC) mutation located on exon 15 were found. Mutations p.Arg778Leu (c.2333G > T) and p.Arg1041Trp (c.3121C > T) located on exons 8 and 14, respectively, were detected with a frequency of 3%. Furthermore, mutations were detected for p.Arg616Gln (c.1847G > A) from exon 6, p.Arg778Gly (c.2332C > G) and p.Trp779* (c.2336G > A) from exon 8, p.Val834Asp (c.2501T > A) from exon 10, p.Gly943Ser (c. 2827G > A) from exon 12, and c.3222_3243 + 21del43 from exon 14 (Figure 1). The remaining mutations were not detected and occurred less frequently than 1:100, that is, the frequency is less than 1%. Based on these mutations, primers for ARMS PCR were designed (Figure 2).

The specificity of ARMS PCR for each mutant primer was conferred by adding an additional mismatch at the 3′ end of the primer. The specificity of ARMS PCR for each mutant primer was conferred by adding an additional mismatch at the 3′ end (Figure 2). Mismatches were selected according to the rule of strong and weak nucleotide mismatches: strong mismatches are C-C, G-A and A-A; and weak mismatches are T-T, T-C, T-G, G-G and A-C [32]. These modifications contribute to the appearance of a “floating tail” in the primer attached to the template DNA and do not allow the polymerase to move further and produce a fragment (Figure 3).

At the development stage of the assay, negative (without DNA samples) and positive (samples without mutation) controls were used, and all data obtained were compared with the results from the Sanger sequencing of all *ATP7B* exons. All data obtained as a result of the ARMS PCR matched the mutant alleles found during the Sanger sequencing. The negative control showed no trace of amplification when tested by electrophoresis and the positive control only showed a reaction with the primer without the mutation. Heterozygous carriers of mutations showed two bands on the agarose gel, which indicated a reaction with the modified and normal primers (Figure 4). All this indicates that the ARMS PCR assays were highly specific for the detection of the presence of a mutation in a homo- or heterozygous state, or to establish the absence of a mutation at a given position. All ARMS PCR experiments were repeated three times to avoid false positive or false negative results.

## 4. Discussion

Patients with WD have a wide range of symptoms associated with various organs. The most common symptoms are liver dysfunction, neuropsychiatric disorders, Kaiser–Fleischer rings on the cornea, and hemolysis caused by acute liver failure. The differences in the rates of accumulation of toxic copper in organs and the individual sensitivity to their damage provide polymorphisms of the clinical manifestations and the age of onset of the disease [5]. An accurate diagnosis is established after a thorough assessment of the clinical picture, biochemical tests, as well as after detecting a violation of copper metabolism. However, even in this case, one should carefully evaluate the data and adjust the results by taking into account the patient’s age, habits, lifestyle, and by not ruling out the possibility of similar diseases [3].

WD belongs to a group of diseases that require diagnosis in the early stages of the disease. The diagnosis of WD is determined by a combination of clinical manifestations and laboratory parameters that indicate a violation of copper metabolism with its accumulation in the liver and brain tissue—a decrease in the content of ceruloplasmin in the blood serum and an increase in the daily excretion of copper in the urine. However, these standard tests can produce both false positive and false negative results early in the disease. Failure to diagnose a patient with WD at an early stage of the disease may lead to the loss of pathogenetic therapy, or inappropriate administration of potentially toxic drugs to such patients with a false positive diagnosis [7,33]. In this regard, an urgent and accurate diagnosis is important when time is critically limited for making vital management decisions. The measurement of serum ceruloplasmin takes half a day but may be falsely normal in patients with acute liver failure. The evaluation of copper in urine requires collection within 24 hours; however, the determination of copper in the liver requires an invasive biopsy, which is potentially dangerous in patients with prolonged coagulation, tissue digestion and copper analysis requiring at least two working days. However, according to the Leipzig Wilson’s disease diagnostic scale, in the presence of two mutations in the *ATP7B* gene, the maximum score immediately occurs, and the diagnosis of WD is established immediately.

Molecular analysis is the most accurate rapid diagnostic tool in both symptomatic and asymptomatic WD patients. However, this method does not allow for the detection of mutations in intron regions, which occur in about 2% of patients with WD [34]. A direct molecular study of WD is recommended due to the diversity of its clinical manifestations and the limitations of modern markers [35]. It also allows for the creation of flexible and efficient mutation screening strategies for certain populations. For example, Lovicu et al. developed multiplex PCR and reverse dot blotting using allele-specific probes to detect six common mutations in Sardinia [36]. Another notable example is Huster et al.; they used multiplex PCR and DNA banding technology to identify four common mutations in Europeans [37]. However, most of the proposed modern methods of molecular diagnostics of BV are still expensive, technically complex and laborious. Most importantly, they are based on an initial study of the spectrum of mutations in the *ATP7B* gene in this population, which also determines the sensitivity of the developed assay itself. If only 1 of the 14 mutations is found, or none of them, then the entire *ATP7B* gene must be sequenced to identify new mutations.

Based on the results from the sequencing of the genetic material of 100 people, 6 people were identified without mutations in the *ATP7B* gene, 27 patients were identified with heterozygous carriage and 67 patients were identified with a pathogenic mutation variant. However, the 6 people that were identified without mutations also had symptoms consistent with WD, including high copper levels in their blood. It is possible that these patients have mutations in the intron regions that prevent proper maturation of the protein, which may also be the cause of WD [19]. Furthermore, among patients with mutations, people with an asymptomatic course of the disease were found. This aspect has been highlighted in articles that indicate that the penetrance of this disease may be below 100% [38,39]. Summing up the above, it is worth noting that our method is well suited as a primary diagnostic tool, but to create a more accurate picture of the patient’s diagnosis, it is also necessary to examine a doctor and conduct a biochemical study.

As a candidate for fast and cheap molecular genetic diagnostics of WD, we proposed allele-specific PCR, which is based on the use of allele-specific primers that are selected strictly for mutation sites. Often, two versions of universal primers can be used to implement the technology: the first option is when the primer must be strictly complementary in its 3′-terminal nucleotide, which corresponds to the nucleotide of the template DNA. The second option is when the universal primer contains a 3′-terminal nucleotide, which is always non-complementary to the template, and the mutant nucleotide of the template falls into its inner part. In this case, there are no PCR products if any non-complementary mutant nucleotide of template DNA, regardless of its exact localization, enters the inner part of the primer in the hybrid. Such primers allow for the detection of any point mutations (SNPs) in the homozygous state. To identify the heterozygous state of the analyzed gene, “mutant” and “normal” primers can be used in two different tubes. SNPdetect polymerase is used for this test.

Our modification consists of designing unique primers that are modified with one nucleotide mismatch at the 3′ end in addition to the mismatch in the mutant primer. Modification occurs according to the rule of strong and weak nucleotide mismatches: strong mismatches are C-C, G-A and A-A; and weak mismatches are T-T, T-C, T-G, G-G, and A-C. This modification allows for the use of Taq DNA polymerase, which reduces the cost of analysis. Using the obtained modified primers, we performed allele-specific PCR with patients who had a spectrum of homo- and heterozygous mutations. The testing of primers and the method was repeated two additional times to eliminate erroneous results, and a negative control was also tested using patients who did not have mutations in the *ATP7B* gene. The results of this analysis are presented in photographs of the gels after gel electrophoresis.

Our method is faster and cheaper than other diagnostic methods, as well as easy to use. The analysis takes about 3 hours from the moment the patient’s blood is received. Compared to Sanger sequencing, our method has significantly fewer steps (DNA extraction, amplification, electrophoresis), which speeds up analysis, reduces the likelihood of error, and reduces the amount of laboratory equipment required [8,9,10]. If we compare this method with NGS, then our method does not require the ability to work with databases, and reagents and equipment for this method are cheaper than equipment for NGS analysis [4]. However, our method is not able to detect new mutations that are not included in the list of the 14 presented alleles. Therefore, this method is best suited for use in small towns as the first line of diagnosis in patients with suspected WD. If the patient does not have mutations among these 14 alleles but has symptoms of WD, then they should contact a larger center with specialists in the field of NGS. In this regard, it is worth noting that the sensitivity of the method is about 94%.

Investment in new equipment and consumables, as well as the need to train employees in new methods, hinders the development and dissemination of high-performance WD diagnostics. However, the cycler is available to most diagnostic laboratories, and the methods developed to detect mutations in the *ATP7B* gene on it do not require great skills in genetics and molecular biology for analysis. Our approach combines the need for even less financial investment and the same high accuracy as other similar methods due to the high specificity of the ARMS primers. Combined with the economical use of the DreamTaq Green PCR Master Mix and electrophoresis reagents, these primers allowed for the specific amplification of target sequences and therefore eliminated the need for costly and complex fluorescently labeled probe synthesis. On the other hand, this method does not allow for the detection of new mutations, and in cases where patients have rare mutations that are not included in the established spectrum, it is necessary to sequence all exons of the *ATP7B* gene.

In conclusion, we described a panel of 14 mutations based on the ARMS assay for mutation screening in patients with WD. This assay has demonstrated good mutation detection accuracy, reproducibility and potential for high-throughput assays in clinical laboratories. This method provides a fast, accurate and direct molecular diagnosis of WD, making conventional diagnostic markers unnecessary, especially in patients with life-threatening fulminant liver failure. In the future, it is necessary to expand the sample of patients to search for new mutations and modify the method to a multiplex method to reduce the cost and speed up the analysis.

## Figures and Tables

**Figure 1 genes-13-01940-f001:**
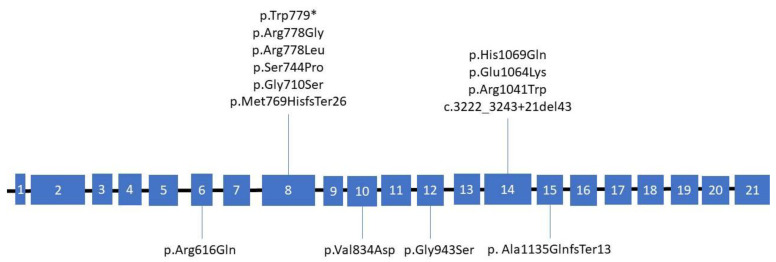
Schematic representation of the exons of the *ATP7B* gene and the location of the detected mutations.

**Figure 2 genes-13-01940-f002:**
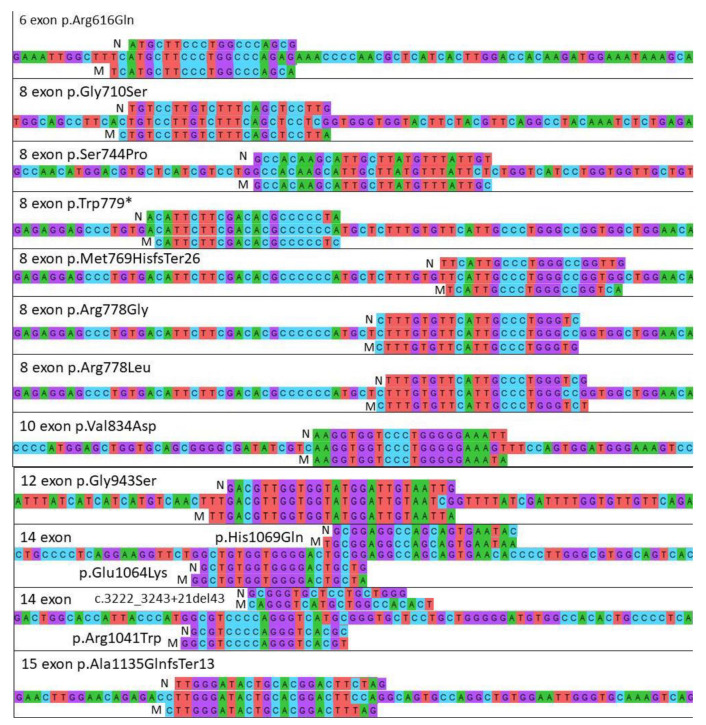
The location of primers for searching for mutations in different exons of the *ATP7B* gene, where N is the primer with the normal allele and M is the primer with the mutant allele.

**Figure 3 genes-13-01940-f003:**
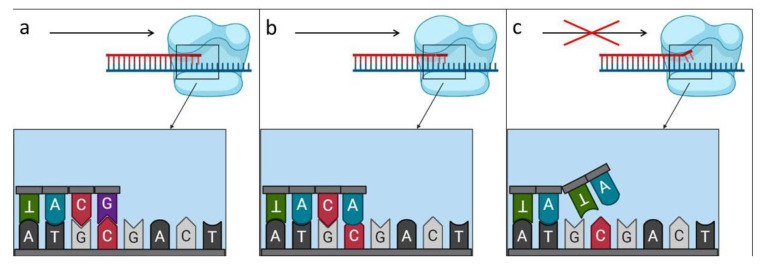
Scheme of attachment of the modified primer and the occurrence of a “floating tail” on the example of: (**a**) no mismatch; (**b**) the presence of one nonconformity; (**c**) the presence of two inconsistencies that prevent the production of the fragment.

**Figure 4 genes-13-01940-f004:**
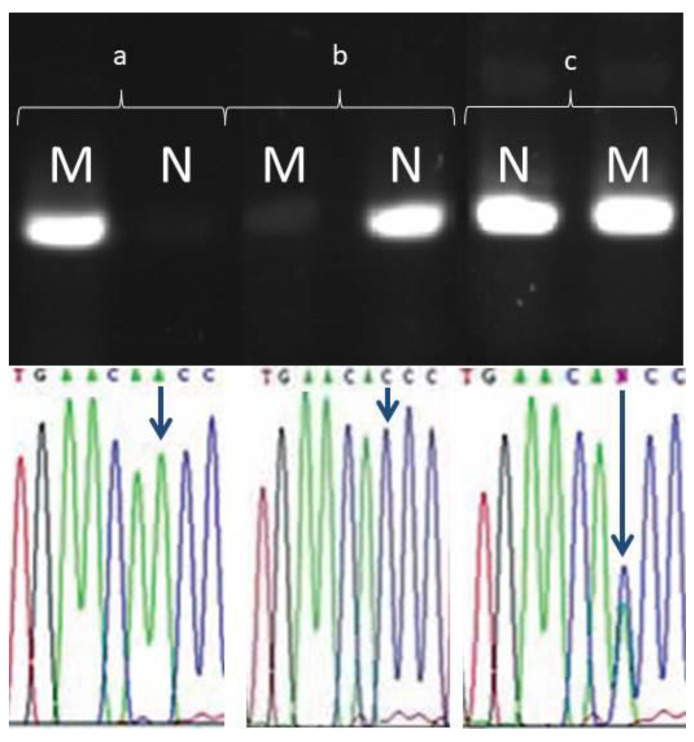
An example of the results from ARMS PCR in patients with His1069Gln (3207C > A) mutation: (a) a mutation in the studied allele, it is in the homozygous state; (b) wild type; (c) a mutation in the studied allele, it is in a heterozygous state.

**Table 1 genes-13-01940-t001:** Primers for exon amplification of the *ATP7B* gene.

Exon	Forward Primer	Revers Primer
2A	GGCATTGTTTTCCATTTTCTCAGTG	CATGTCCCCAATTTGATGGCAAAC
2B	GAAGGTTTCCCTGGAACAAGGC	GCAGAAGATAAAGGTCTCTTTGGG
2C	TGGGACCAATTGATATTGAGCGG	GGAAGACCTGTGATCTGTCCC
2D	TATCGAGGCACTTCCACCTGG	CTCACCTATACCACCATCCAGG
3	GGTGGGAGCCGGGACAATGAACC	CAGCATTCCTAAGTTCAACATGGG
4	TATTGACTGTGTCAACCTAGAGGC	AAACTGTCAGAAGCCTGTAACCC
5	CTGCCTGTTACCTAGACTCCC	TTACCCATTCACTGATATCCTCCC
6	AAAACCCACAAAGTCTACTGAGGC	CAGCTAATCCAGGAGGAAGGC
7	TCTTAAACTGTGTCCTCAGAAGGG	ACTATGTTTGCGCTTAGCGGGC
8	GACTGTGCACAAAGCTAGAGGC	GAGATTTGTTTACTGAAGGAGCAGC
9	GTGTGGTGGATAGCAAGTAACGC	CTTTCGTAGCTGGATTGAGAGTGG
10 + 11	AGCTGGCCTAGAACCTGACCC	GAACTCTTCACATAATTTCTAAAACGAGA
12	CCCAATCTTTATCCATGCTTGTGG	AGTGACTGTTTATCCTACTCTGGC
13	CCTTATTGAACTCTCAACCTGCC	CTCTGTTGCTACTGTTGTTATTCCC
14	CCCTGAGATTGAACGACAGAGG	GGTGAGGAATAAAAGAGCATTGGC
15	CTTTCCGCTGCTCTCTTGCC	CAGAGGCAATCACTGCTGGG
16	TGTCACAAGAGGTGCTTACAAGG	GCCTGAAATTAAGAGAGGAAGGC
17	CTTCCAGACTTTTGTGTACATCCG	GAGTACAGCTCAGTGCTGGG
18 + 19	TTTTGCCAACACTAGGCATTGCC	GGAGACAGAAGCCTTTCTGGG
20	GAACATCAGGGCGAGTGGAAG	GTGCTAAGCATGCAGAATGACAAG

**Table 2 genes-13-01940-t002:** Primers for allele-specific PCR of the *ATP7B* mutation sites.

Exon	Mutation	Mutant forward Primer	Normal forward Primer	Revers Primer
5	Arg616Gln	TTTGACCCGGAAATTATCGGTCCATA	GACCCGGAAATTATCGGTCCATG	CAAGCTCTCCACAACAAGAGTGG
8	Met769HisfsTer26	CATTCTTCGACACGCCCCCTC	ACATTCTTCGACACGCCCCCTA	TTTGGAGATTAGTGACTAGAGCACCT
	Gly710Ser	CTGTCCTTGTCTTTCAGCTCCTTA	CTGTCCTTGTCTTTCAGCTCCTTG	
	Ser744Pro	GGCCACAAGCATTGCTTATGTTTTTC	TGGCCACAAGCATTGCTTATGTTTTTT	
	Arg778Leu	CTTTGTGTTCATTGCCCTGGGTCT	TTTGTGTTCATTGCCCTGGGTCG	GACCAACTACATATTCAGTTTTGCACC
	Arg778Gly	CTTTGTGTTCATTGCCCTGGGTG	CTTTGTGTTCATTGCCCTGGGTC	
	Trp779*	TCATTGCCCTGGGCCGGTCA	TTCATTGCCCTGGGCCGGTTG	
10	Val834Asp	GGGGCGATATCGTCAAGGTGTA	GGGGCGATATCGTCAAGGTGTT	GTCTGATTTCCCAGAACTCTTCACATA
12	Gly943Ser	TTGACGTTGGTGGTATGGATTGTAATTA	GACGTTGGTGGTATGGATTGTAATTG	CAACTGAGCACCAATTGGTGTCTG
14	His1069Gln	TGCGGAGGCCAGCAGTGAATAA	GCGGAGGCCAGCAGTGAATAC	AGACTGCCCGTACTCCCCAAG
	Glu1064Lys	GGCTGTGGTGGGGACTGCTA	GCTGTGGTGGGGACTGCTG	
	Arg1041Trp	GGCGTCCCCAGGGTCACGT	GCGTCCCCAGGGTCACGC	
	3222_3243+21del43	GGCGTGGCGTACGTGGACTT	GTGGCAGTCACCAAATACTGTAAAG	
15	Ala1135GlnfsTer13	TGAGGCTGGCAGCCTTCCCTA	AGGCTGGCAGCCTTCCCTC	ACATTTCCCTCTGCTTTCCCTGAC

## Data Availability

Not applicable.

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
