# Peer review of "Design, Optimization and Validation of the ARMS PCR Protocol for the Rapid Diagnosis of Wilson’s Disease Using a Panel of 14 Common Mutations for the European Population"

_genes, 2022, doi:10.3390/genes13111940_

Round 1

Reviewer 1 Report

The most prevalent variants in Russia are p.His1069Gln, p.Met769HisfsTer26 and p.Ala1135GlnfsTer13, but what are the criteria to select the other ones (until 14)

Lines 209-216 are not results are methods

Were the 100 samples previously tested by Sanger considered the gold standard method? In line 223 the authors said and all data obtained were com-223 pared with the results of Sanger sequencing of all ATP7B exons but no data are shown comparing the results between the two methods.

The concordance results from sanger are the same that authors describe with the ARMS PCR?

What are TAT using ARMS PCR comparing with other methodologies?

Advantage, disadvantage related to NGS

duplication and deletion of exons of the ATP7B gene are rare and contribute to a very small number of WD cases that are missed with this method

Author Response

Good afternoon dear reviewer. We are very glad to receive your comments and remarks as this is a new topic for our laboratory. Thank you very much for your comments! Corrected version attached.
1) The most prevalent variants in Russia are p.His1069Gln, p.Met769HisfsTer26 and p.Ala1135GlnfsTer13, but what are the criteria to select the other ones (until 14)
1) Thank you for this comment, we decided to write a clarifying paragraph on what parameters the mutations were chosen for, it starts from line 253.
2) Lines 209-216 are not results are methods
2) We apologize, we did not quite understand your remark, since these lines refer to chapter 2.2. methods.
3) Were the 100 samples previously tested by Sanger considered the gold standard method? In line 223 the authors said and all data obtained were com-223 pared with the results of Sanger sequencing of all ATP7B exons but no data are shown comparing the results between the two methods.
3) Thank you for your comment, we also indicated the answer to this question in the paragraph that begins on line 253. 
4) The concordance results from sanger are the same that authors describe with the ARMS PCR?
4) Thank you for this comment, clarified and added on line 298.
5) What are TAT using ARMS PCR comparing with other methodologies?
5) Thank you for this comment, clarified and added on line 393.
6) Advantage, disadvantage related to NGS
6) This question is also described in the paragraph starting with the line 393.
7) duplication and deletion of exons of the ATP7B gene are rare and contribute to a very small number of WD cases that are missed with this method
7) We apologize, but we did not really understand this question. We agree with you that large deletions and duplications are poorly detected by this method and are quite rare.
Thank you again for your comments. Best regards, Garbuz M.M.

Reviewer 2 Report

It is very well written paper according to diagnosis of WD (genetic tests) I have few comments, the frequency of the p.His1069Gln was the  highest in Western Europe than described in article  - see the paper Gromadzka, et al. Mov Disord 2006 - "The p.His1069Gln was the most common mutation (allelic frequency: 72%)" , so it is worth to mention in introduction or in disuccsion thats in some regions , countries the analysis of few mutation can covered >80% of mutations (in Poland panel of 4 mutations e.g.).

Another issue of WD diagnosis is to be carefull for diagnosis of WD alone on genetic tests - it is worth in study limitation mention, that's diagnosis is more complex . see the paper  

Pitfalls in diagnosing Wilson's Disease by genetic testing alone: the case of a 47-year-old woman with two pathogenic variants of the ATP7B gene.

Antos A, Litwin T, Skowrońska M, Kurkowska-Jastrzębska I, Członkowska A.Neurol Neurochir Pol. 2020;54(5):478-480. doi: 10.5603/PJNNS.a2020.0063. 

Netherless the paper is very interesting, and after correction is worth to publish

Author Response

Good afternoon dear reviewer. We are very glad to receive your comments and remarks as this is a new topic for our laboratory. Thank you very much for your comments! Corrected version attached.

1) It is very well written paper according to diagnosis of WD (genetic tests) I have few comments, the frequency of the p.His1069Gln was the  highest in Western Europe than described in article  - see the paper Gromadzka, et al. Mov Disord 2006 - "The p.His1069Gln was the most common mutation (allelic frequency: 72%)" , so it is worth to mention in introduction or in disuccsion thats in some regions , countries the analysis of few mutation can covered >80% of mutations (in Poland panel of 4 mutations e.g.).
1) Thank you for this comment, we are happy to supplement our article with the information you suggested. This paragraph starts on line 76.

2) Another issue of WD diagnosis is to be carefull for diagnosis of WD alone on genetic tests - it is worth in study limitation mention, that's diagnosis is more complex . see the paper  Pitfalls in diagnosing Wilson's Disease by genetic testing alone: the case of a 47-year-old woman with two pathogenic variants of the ATP7B gene. Antos A, Litwin T, Skowrońska M, Kurkowska-Jastrzębska I, Członkowska A.Neurol Neurochir Pol. 2020;54(5):478-480. doi: 10.5603/PJNNS.a2020.0063. 

2) Thank you so much for this comment, we also gladly supplemented our article with the information that you suggested and considered this issue more widely than before. These paragraphs begin on lines 358 and 393.

Thank you again for your comments. Best regards, Garbuz M.M.

Reviewer 3 Report

The authors report a method to identify the 14 most common  mutations in Wilson Disease (WD) in their region and suggest this could be useful as a fast way to diagnose WD. The quality of english is very good.

I do not have the qualifications to comment on the specific analytical methodology  and suggest the Journal also include a reviewer with specific insight in the specific methodological issues.

A methodologically experienced reviewer should also judge if the methodological aspects can be shortened (as I would expect).

Apart from that, I have some concerns that authors should address.

Major concerns

1)      Lines 84-98. Lines from the Journals author instruction that should have been deleted before submission. 

2)      The authors argue that their methodology is faster that current methodologies. But they do not specify the time consume of their methodology, and should do so.

3)      It is unclear what the patient material really were. Were they all patients with suspected WD ? How many were finally diagnosed with WD. Maybe some were already know with WD - How many had already been diagnosed with WD ? , How many WD patients had only one or no mutations detected.

4)      Is it possible to calculate a sensitivity and specificity of the method (depends on the patient material).

5)      There should be a discussion of how often WD patients will have other mutations than those detected by their method.  

6)      The authors should address the possibility that penetrance is lower than 100% as suggested by recent publications (e.i. Sandahl TS et al.  Hepatology 2020; (71):p 722). That will affect the use of their method as a first choise diagnostic tool. In my view, measurements of copper metabolism are still necessary. 

Minors

Line 41. Provide a reference for 1:30,000-45,000 (could be Sandahl et al mentioned above)

Line 260. Normal ceruloplasmin can be seen in non-acute WD patients. In acute liver failure it will usually be low but that is also the case in many other patients with non-WD  acute liver failure. Heterozygotes may have falsely low ceruloplasmin.

Line 276. There is a reference to ref 22, but that is missing in the reference list.

Author Response

Good afternoon dear reviewer. We are very glad to receive your comments and remarks as this is a new topic for our laboratory. Thank you very much for your comments! Corrected version attached.

Major concerns

1)      Lines 84-98. Lines from the Journals author instruction that should have been deleted before submission. 
1) Thank you very much for this comment, we apologize for this oversight. Fixed in new version.

2)      The authors argue that their methodology is faster that current methodologies. But they do not specify the time consume of their methodology, and should do so.
2) Thank you for this comment. We are pleased to describe this question more broadly, the answer to it begins on line 393.

3)      It is unclear what the patient material really were. Were they all patients with suspected WD ? How many were finally diagnosed with WD. Maybe some were already know with WD - How many had already been diagnosed with WD ? , How many WD patients had only one or no mutations detected.
3) Thank you for this comment. We are happy to rewrite this section to better describe the experiment. The paragraph from line 358 was also written to supplement the results.

4)      Is it possible to calculate a sensitivity and specificity of the method (depends on the patient material).
4) Thank you for this comment. We are happy to add content. It is located in a new paragraph on line 404.

5)      There should be a discussion of how often WD patients will have other mutations than those detected by their method.  

5) 

6)      The authors should address the possibility that penetrance is lower than 100% as suggested by recent publications (e.i. Sandahl TS et al.  Hepatology 2020; (71):p 722). That will affect the use of their method as a first choise diagnostic tool. In my view, measurements of copper metabolism are still necessary. 

6) Thank you for this comment. We have written a new paragraph to complement the experiment. It starts on line 253.

Minors

1) Line 41. Provide a reference for 1:30,000-45,000 (could be Sandahl et al mentioned above)
1) Thank you for this comment. Fixed in new version.

2) Line 260. Normal ceruloplasmin can be seen in non-acute WD patients. In acute liver failure it will usually be low but that is also the case in many other patients with non-WD  acute liver failure. Heterozygotes may have falsely low ceruloplasmin.

2) Thank you for this comment. We described this information in the paragraph that starts from line 358.

3)Line 276. There is a reference to ref 22, but that is missing in the reference list. 3) Thank you very much for this comment, we apologize for this oversight. Fixed in new version.

Thank you again for your comments. Best regards, Garbuz M.M.

Round 2

Reviewer 1 Report

no comments